# Influence of Sepsis on the Middle-Term Outcomes for Urinary Tract Infections in Elderly People

**DOI:** 10.3390/microorganisms11081959

**Published:** 2023-07-31

**Authors:** Arturo Artero, Ian López-Cruz, Juan Alberola, José María Eiros, Elena Resa, Laura Piles, Manuel Madrazo

**Affiliations:** 1Universidad de Valencia, 46017 Valencia, Spain; arturo.artero@uv.es; 2Hospital Universitario Doctor Peset, 46017 Valencia, Spain; ilopezcruz5@gmail.com (I.L.-C.); eiros@med.uva.es (J.M.E.); elenaresarui@gmail.com (E.R.); laurapilesroger@gmail.com (L.P.); manel.madrazo@gmail.com (M.M.)

**Keywords:** sepsis, outcomes, urinary tract infection (UTI), elderly

## Abstract

Urinary tract infection (UTI) is a common condition that predominantly affects elderly people, who are particularly susceptible to developing sepsis. Previous studies have indicated a detrimental effect of sepsis on short-term outcomes in elderly patients with UTI, but there is a lack of data about the middle-term prognosis. The aim of this study was to investigate the influence of sepsis on the middle-term prognosis of patients aged 65 years or older with complicated community-acquired UTIs. A prospective observational study of patients admitted to a hospital with UTI. We conducted a comparison of epidemiological and clinical variables between septic and nonseptic patients with UTI, as well as their 6-month case-fatality rate. A total of 412 cases were included, 47.8% of them with sepsis. Septic patients were older (83 vs. 80 years, *p* < 0.001), but did not have more comorbidities. The short-term case-fatality rate was higher in septic patients and this difference persisted at 6 months (34% vs. 18.6%, *p* = 0.003). Furthermore, age older than 75 years, Barthel index <40 and healthcare-associated UTI were also associated with the middle-term case-fatality rate. In conclusion, the detrimental impact of sepsis is maintained on the middle-term prognosis of elderly patients with UTI. Age, functional status and healthcare-associated UTIs also play significant roles in shaping patient outcomes.

## 1. Introduction

Urinary tract infections (UTIs) are a prevalent condition affecting individuals of all ages, with particular significance among elderly patients [1]. UTI mostly occurs in women, but its incidence increases with age in men, and men older than 80 years have a similar infection rate to women [2]. With age, the risk factors for UTIs multiply, encompassing anatomical abnormalities and compromised immune responses [1]. Consequently, elderly individuals are more susceptible to developing complicated UTIs, which can result in severe consequences [3].

Sepsis, a life-threatening organ dysfunction caused by a dysregulated host response to infection, is a common disorder in elderly patients with complicated UTIs [4]. The presence of sepsis can lead to a cascade of physiological changes, contributing to organ dysfunction and increased mortality [5]. In the United States, sepsis has been found to contribute to every two-to-three hospital deaths [6]. Elderly individuals are particularly susceptible to sepsis due to age-related immunosenescence and comorbidities that compromise their immune response [7,8]. The development of sepsis in the context of a complicated UTI further exacerbates the already delicate health status of these patients. Some studies found higher long-term mortality among survivors of an episode of sepsis from any source [9,10], as well as loss of their functional status, quality of life or cognitive capabilities [11,12,13,14]. The mechanisms of this phenomenon have not been well clarified.

Identifying the site of infection may be critical for the prognosis. The Surviving Sepsis Campaign (SSC) advises immediate evaluation of patients with severe sepsis or septic shock to identify potential sites of infection suitable for source control. Once diagnosed, it is recommended that source control procedures be performed within 12 h [15]. However, the precise effect of timing on source control in sepsis patients remains uncertain, and further investigation is required.

Although urinary sepsis has been considered to have a better prognosis than sepsis from other sources [16,17], several studies have demonstrated the detrimental effect of sepsis on the short-term outcomes of elderly patients with UTIs [18,19,20]. However, there is a lack of data about middle-term prognosis in this specific population of the study.

Examining the middle-term outcomes of elderly patients admitted to hospitals with complicated UTIs, with or without sepsis, is essential for enhancing clinical management strategies and optimizing patient care. The aim of this study was to investigate the influence of sepsis on the medium-term prognosis, as assessed by the 6-month case-fatality rate of a series of patients aged 65 years or older admitted to a hospital with complicated community-acquired UTI.

## 2. Material and Methods

### 2.1. Study Design and Patients

Cohort prospective observational study of patients aged 65 years or older consecutively admitted to an internal medicine ward at a university hospital, diagnosed with complicated community-acquired UTI, from January 2017 to December 2021. The diagnosis of UTI was initially created by the attending physician in the hospital emergency department and subsequently confirmed by the physician in charge after admission. This confirmation was based on a thorough assessment of the patient’s medical history, physical examination, as well as laboratory and microbiological diagnostic procedures. Patients were categorized into two groups based on the presence or absence of sepsis. The primary outcome measure was a 6-month case-fatality rate. Cases with a negative urine culture or a clinical syndrome compatible with any other condition, after being reviewed by two independent researchers, were excluded, as well as nosocomial or UTI cases transferred from the intensive care unit (ICU). Standard clinical care for UTI and sepsis was implemented without any modifications in its management. The flowchart of the inclusion of 412 cases of complicated community-acquired urinary sepsis is shown in Figure 1.

This study was approved by the Clinical Research Ethics Committee of the Doctor Peset University Hospital (code 85/16, September 2016) and followed the STROBE statement. All patients provided their signed consent to participate in the study.

### 2.2. Data Collection and Definitions

The researchers extracted epidemiological and clinical variables from the electronic medical record using a predefined data collection form [21]. Patient follow-up was conducted for up to six months after discharge, during which time their prognosis and outcomes were recorded.

The definition of a complicated urinary tract infection involved the identification of pyuria and bacteriuria through urinalysis using microscopy. Additionally, at least one of the following criteria needed to be met: (i) presence of cystitis symptoms (such as dysuria, urinary urgency and/or urinary frequency) accompanied by fever or other signs of systemic illness, such as chills, rigors or acute mental status changes; (ii) experiencing flank pain and/or costovertebral tenderness; (iii) in the absence of localizing symptoms and after ruling out other potential causes, fever or sepsis were considered indications of complicated urinary tract infection. Community-onset healthcare-associated UTI (HCA-UTI) was defined as a urinary tract infection that originated in the community and met any of the following criteria: (I) hospital admission within the 90 days prior to the current hospital admission; (II) receiving antimicrobial therapy within the 90 days prior to the current hospital admission; or (III) residence in a nursing home [22]. Community-acquired infection was defined as a urinary infection in which symptoms originated in the community and did not meet any of the aforementioned criteria [22].

Urinary sepsis was diagnosed according to sepsis-3 criteria [5] and identified as an acute change in total SOFA score ≥ 2 points consequent to the infection. SOFA and quick SOFA (qSOFA) scales were used according to their original definitions [23]. The acute physiology and chronic health evaluation classification system (APACHE II) score was used to identify illness severity at admission [24]. The Barthel index was used to evaluate the functional status of the patients [25].

Clinical symptoms and signs were directly obtained through patient interviews and physical examinations. Comorbidities included were diabetes mellitus (determined by fasting plasma glucose values ≥ 126 mg/dL or glycated hemoglobin values ≥ 6.5 percent), cognitive impairment (based on clinical criteria for dementia or mild neurocognitive disorder as defined in the Diagnostic and Statistical Manual for Mental Disorders, Fifth Edition [DSM-5]), chronic kidney disease (defined by a decreased glomerular filtration rate of less than 60 mL/min according to CKD-EPI equation), chronic obstructive pulmonary disease (COPD, confirmed by spirometry showing airflow limitation with a forced expiratory volume in one second/forced vital capacity [FEV1/FVC] ratio less than 0.7 or lower than the lower limit of normal, which is not completely reversible after the administration of an inhaled bronchodilator), active or previous cancer and indwelling urinary catheter. Fever was considered present if the patient reported a temperature of ≥38 °C at home or if it was measured in the emergency department. Laboratory assessments included coagulation testing, a complete blood count and blood chemistry analysis (including liver and renal function evaluation and electrolyte measures, procalcitonin and C-reactive protein levels).

The Charlson comorbidity index was used as a measure of total comorbidity burden, considering a Charlson comorbidity index equal to or greater than 3 as indicative of significant comorbidity. The score assigned to each subgroup of diseases was as follows: 6 points for AIDS/HIV, metastasic solid tumor and any tumor or malignancy (including metastasic disease); 3 points for diabetes with end-organ damage and moderate or severe liver disease; 2 points for diabetes with complications, moderate or severe renal disease, hemiplegia or paraplegia, leukemia or lymphoma; and 1 point for myocardial infarction, congestive heart failure, peripheral vascular disease, cerebrovascular disease, dementia, chronic pulmonary disease, connective tissue disease (including rheumatoid arthritis, systemic lupus erythematosus, polymyalgia rheumatica, dermatomyositis/polymyositis or mixed connective tissue disease, based on their clinical criteria), peptic ulcer disease and mild liver disease [26].

Inadequate empirical antimicrobial therapy (IEAT) was considered the occurrence of an infection that was not effectively treated at the time when the causative micro-organism and its antibiotic susceptibility were known. This included the absence of antimicrobial agents directed at a specific class of micro-organisms, and the administration of an antimicrobial agent to which the micro-organism responsible for the infection was resistant [27]. Multidrug-resistance bacteria (MDR-B) was defined according to an international expert proposal by Magiorakos et al. [28], as nonsusceptibility to at least one agent in three or more antimicrobial categories (extended-spectrum penicillins, carbapenems, cephalosporins, aminoglycosides, and fluoroquinolones for gram-negative bacteria; and ampicillin, vancomycin, fluoroquinolones, fosfomycin and linezolid for gram-positive bacteria).

Microbiological data were collected through urine and blood cultures, as well as susceptibility testing. This encompassed the identification of bacteremia, the determination of the causative agents of UTI through culture isolation, the assessment of resistance patterns exhibited by the isolated micro-organisms, and identification of cases involving polymicrobial infections. The microbial identifications of the urine cultures were created using the Bruker MALDI Biotyper system (Beckman Coulter, Brea, CA, USA), and for the drug sensitivity and resistance tests, the DxM MicroScanWalkAway microbiology system (Beckman Coulter, Brea, CA, USA) was used. This is a microbroth dilution method based on a combination of CLSI and EUCAST rules. Blood cultures were obtained in the emergency department and were processed using the BacT/ALERT^®®^ VIRTUO™, which is an automated system for the culture and detection of micro-organisms in blood. Micro-organisms isolated from positive blood cultures were identified using the Bruker MALDI Biotyper system (Beckman Coulter, Brea, CA, USA) and the drug sensitivity and resistance tests. The DxM MicroScan WalkAway microbiology system (Beckman Coulter, Brea, CA, USA), and the VITEK 2 system (Biomerieux) are two systems using microbroth dilution methods according to a combination of CLSi and EUCAST rules.

### 2.3. Statistical Analysis

Quantitative variables were compared using the Mann–Whitney U-test. Qualitative variables were compared with the chi-square test and Fisher’s exact test. Kaplan–Meier method was used to compare survival rates between patients with and without sepsis. Cox proportional hazards regression was used to evaluate the significance of the observed differences between the survival curves. Multivariate analysis was performed using logistic regression, considering an α significance level of 0.05 for all tests. The statistical package SPSS version 22 from IBM for Windows was used for the statistical analysis.

This study was approved by the Clinical Research Ethics Committee of the Doctor Peset University Hospital (code 85/16, September 2016) and followed the STROBE statement. All patients provided their signed consent to participate in the study.

## 3. Results

Out of the 1198 cases of UTI diagnosed at admission, 412 cases were analyzed, 47.8% of them with sepsis. The median age was 81 years, with 97.6% of the patients having high comorbidity (Charlson ≥ 3), and genders were similarly distributed (50.7% females). Diabetes mellitus, chronic kidney disease and dementia (33%) were the more common comorbidities (see Table 1).

*Escherichia coli* (52.5% in total, with 51.9% and 56.9% in septic and nonseptic patients, respectively; *p*=0.649), *Klebsiella pneumoniae* (12.3% in total, with 16.8% and 10.7% in septic and nonseptic patients, respectively; *p* = 0.073), *Enterococcus faecalis* (9.1% in total, with 8.1% and 11.6% in septic and nonseptic patients, respectively; *p* = 0.235) and *Pseudomonas aeruginosa* (7.5% in total, with 8.6% and 7% in septic and nonseptic patients, respectively; *p* = 0.326) were the most common micro-organisms isolated in the urine cultures, with no differences between septic and nonseptic patients; there were also no differences in UTI due to multidrug-resistant bacteria (35.5% vs. 36.3%, in septic and nonseptic patients, respectively; *p* = 0.875), polymicrobial UTI (11.2% vs. 8.4%, in septic and nonseptic patients, respectively; *p* = 0.338) or IEAT (24.4% vs. 27.6%, in septic and nonseptic patients, respectively; *p* = 0.460). In patients with a urinary catheter, the micro-organisms identified in the urine culture were as follows: *Escherichia coli* (35.6%), *Klebsiella pneumonia* (17.2%), *Pseudomonas aeruginosa* (25.3%), *Proteus mirabilis* (3.4%) and *Enterococcus faecalis* (19.5%). Both *Pseudomonas* and *Enterococcus* were more prevalent in patients with a urinary catheter compared to patients without urinary catheterization.

Sepsis was associated with age (83 years [77–89] vs. 80 years [74–85], *p* < 0.001 in septic and nonseptic patients, respectively), but it was not associated with sex. Septic patients, as might be expected, presented positive qSOFA (≥2 points) and septic shock in more cases (Table 1). The median APACHE II score was 16 in septic patients and 10 in nonseptic patients (*p* < 0.001). Several clinical characteristics, such as altered mental status (65.3% vs. 28.4% in septic and nonseptic patients, respectively, *p* < 0.001), respiratory rate ≥ 22 bpm (39.8% vs. 6.5, in septic and nonseptic patients, respectively, *p* < 0.001) and systolic blood pressure < 100 mmHg (34.7% vs. 2.8%, in septic and nonseptic patients, respectively, *p* < 0.001), were more common in septic patients. However, acute pyelonephritis was more frequent in nonseptic patients (55.8% vs. 70.7%, *p* = 0.002), and fever was frequent in both septic and nonseptic patients without any statistically significant differences.

The in-hospital case-fatality rate was 9.5%, and the post-discharge case-fatality rate increased over time to 26% at 6 months (Table 2). In-hospital case-fatality rates were higher in septic patients (18.3% vs. 1.4%, *p* < 0.001), as well as 30-day case-fatality rates (23.9% vs. 4.7%, *p* < 0.001) and 6-month case-fatality rates (34% vs. 18.6%, *p* = 0.003). The higher case-fatality rate observed in septic patients was maintained over the 6-month follow-up (Figure 2), and the average survival time for patients with and without sepsis was 24 ± 13 days and 132 ± 13 days (*p* = 0.007), respectively. Furthermore, patients with sepsis had a longer hospital stay (6 [4,5,6,7,8] vs. 5 [3,4,5,6,7] days, *p* < 0.001; in septic and nonseptic patients, respectively).

The antibiotics are more frequently empirically used in patients who died, and those who did not die within 6 months of the follow-up were ceftriaxone (43 vs. 47.5%), meropenem (23.4 vs. 18.4%) and quinolones (8.4 vs. 4.9%), with no statistically significant differences between groups. From these antibiotics, only meropenem showed a difference in IEAT (13.1 vs. 22%, *p* = 0.013), while ceftriaxone (47.7 vs. 45.9%) and quinolones (7.4% vs. 5.3%) had no statistically significant differences.

Risk factors for 6-month case-fatality rates were analyzed, choosing among known risk factors for mortality, such as age, comorbidity [29], dependency [30], healthcare-associated UTI, sepsis, lactate ≥ 2 mg/dL [31] and IEAT [19], which were all statistically significant in the univariate analysis. Age older than 75 years (aOR 2.5, 95% CI 1.1–5.6), severe dependency (Barthel < 40) (aOR 5.1, 95% CI 3–8.7), healthcare-associated UTI (aOR 1.7, 95% CI 1.1–2.9) and sepsis (aOR 1.9, 95% CI 1.1–3.1) were associated with the middle-term case-fatality rate (Table 3), while fever seemed to be a protective factor (aOR 0.5 (0.3–0.9), *p* = 0.018). Interestingly, comorbidity or IEAT were not related to the middle-term case-fatality rate.

## 4. Discussion

The results of this study provide evidence that sepsis in elderly patients with UTI has lasting effects beyond the hospitalization period, leading to a worse medium-term prognosis, as indicated by a higher case-fatality rate at 6 months. Additionally, age, functional status, as measured by the Barthel index, and healthcare-associated UTI were identified as important factors influencing patient outcomes. Understanding these factors is crucial for improving post-discharge care and optimizing patient outcomes.

There were no differences in microbiological factors, such as micro-organisms cultured in urine and blood, or the appropriateness of empirical antimicrobial therapy between septic and nonseptic patients. While these factors would indeed have a greater influence on short-term prognosis [19], it is important to note that, in this study, they were not associated with the mid-term case-fatality rate.

The use of the 28-day case-fatality rate as an endpoint for clinical studies may lead to inaccurate inferences [11]. In this study, the 30-day case-fatality rate in septic patients was 23.9%, higher than 4% in one study with older patients with UTI at the Emergency Department, but lower than in older studies with severe sepsis from different sources (35.8–42.5%) [32,33,34]. However, that difference is not so evident at mid-term (6 months), being 34% in septic patients in this study, close to the range of those other older studies (39.4–54.6%) [33,34]. In-hospital mortality and short-term mortality in sepsis have decreased in the last 20 years [11], however, mid- and long-term mortality and disability at 6 and 12 months seem to remain similar [11,35], as sepsis exerts an independent and potentially causal effect on post-acute mortality [23].

The Barthel index was used to measure the individual’s functional independence and ability to perform activities of daily living [25]. In this study, a Barthel index lower than 40 was associated with a 6-month case-fatality rate, similar to what has been previously described in other studies on sepsis from various sources [18,30,36] using the Barthel index or other dependency scores. The true answer to whether one treatment is better than another in sepsis lies in the person’s trajectory in the 6 months to 1 year following the infection, and not only in the 4-week outcome [37]. Some of these differences may be related to the fact that in this study the cohort was older than that in other studies (81 [75–88] years versus 75.8 ± 7.5 to 76.9 ± 8.8 years) [12,38].

Our findings show that age is an independent risk factor for mortality in complicated UTIs, both with and without sepsis, which is consistent with many other previous studies on sepsis from different sources [5,39,40]. In fact, a significant number of sepsis-related deaths in developed countries occur among older, frail patients nearing the end of their lives [41]. In a systematic review, Shankar-Hari et al. found an additional 16.1% of deaths among patients with acute sepsis occurring within one year after hospital discharge in those who survived the initial sepsis episode (referred to as post-acute mortality). Common predictors of post-acute mortality in most of the studies included were age and comorbidity. However, the authors highlighted that the studies conducted were hindered by limited access to comprehensive information on comorbidity, as well as the omission of several crucial factors, such as the trajectory of prior hospitalizations, nutritional status, discharge location and family support [39]. In our study, we did not find an association between the Charlson comorbidity index equal to or higher than 3 points and the middle-term case-fatality rate. This could be explained by a greater influence of poor functional status on mortality, which could more accurately reflect the burden of comorbidities on the patient and their life expectancy. Given the observed effects of confounding, it is reasonable to assume that future research will need to obtain more precise information on confounding factors that would further diminish any independent association between sepsis and post-acute mortality. Consequently, when evaluating sepsis mortality, epidemiological and clinical factors, aside from prompt source control and timely antibiotic administration, should be considered to achieve optimal sepsis care.

Another finding from this study was the significant association between healthcare-associated UTIs and an increased mid-term case-fatality rate. Contrary to the recommendations for pneumonia, in which the classification as “healthcare-associated pneumonia” is discouraged due to the potential risks of inappropriate use of broad-spectrum antibiotics without clear benefits in identifying more severe patients [42,43], healthcare-associated UTI can be useful due to its association with multidrug-resistant bacteria, IEAT and sepsis [16,19,44,45]. The data from our study show that, in addition to a negative short-term impact, healthcare-associated infection is associated with a worse prognosis, including a higher 6-month case-fatality rate. The cause of this phenomenon may be multifactorial, considering the greater comorbidity and worse baseline status of patients with this kind of infection.

These findings emphasize the need for tailored post-discharge care to address the specific needs of elderly patients with complicated UTIs, particularly those who experienced sepsis [12,38]. Strategies should focus on infection prevention, early detection and management of sepsis and comprehensive rehabilitation programs to improve functional status, as recommended in other studies [11,35,39].

The present study has several limitations. Firstly, it was conducted at a single center, which may limit the generalizability of the findings to a broader population. Secondly, this study focused solely on the 6-month case-fatality rate without considering other clinical outcomes such as cognitive impairment or the performance status of the patients. Despite these limitations, the study provides valuable insights into the influence of sepsis on middle-term outcomes with the strength of rigorous patient selection, all of whom had urinary tract infections as the sole source of sepsis.

## 5. Conclusions

In conclusion, our study highlights the detrimental impact of sepsis on medium-term prognosis, as reflected by the increased case-fatality rate at 6 months in elderly patients with complicated UTI. Age, functional status and healthcare-associated UTIs also play significant roles in shaping patient outcomes. These findings underscore the importance of implementing targeted interventions and improving post-discharge care to optimize patient outcomes. Future research should further investigate the specific mechanisms underlying the observed associations and evaluate the effectiveness of tailored interventions in improving outcomes for elderly patients with UTI and sepsis.

## Figures and Tables

**Figure 1 microorganisms-11-01959-f001:**
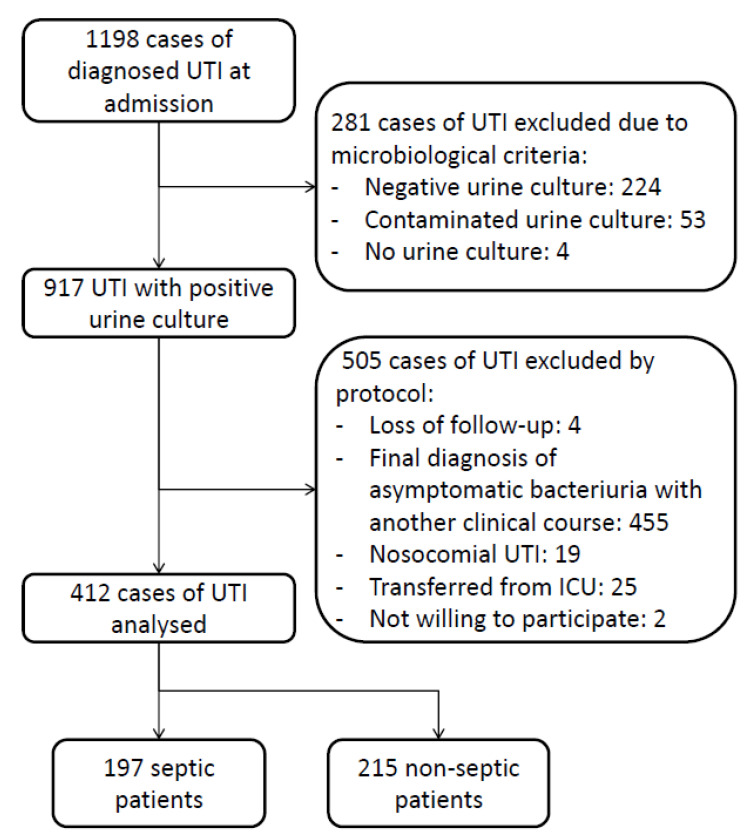
Flowchart of inclusion of 412 patients aged 65 years or older with complicated community-acquired urinary tract infections.

**Figure 2 microorganisms-11-01959-f002:**
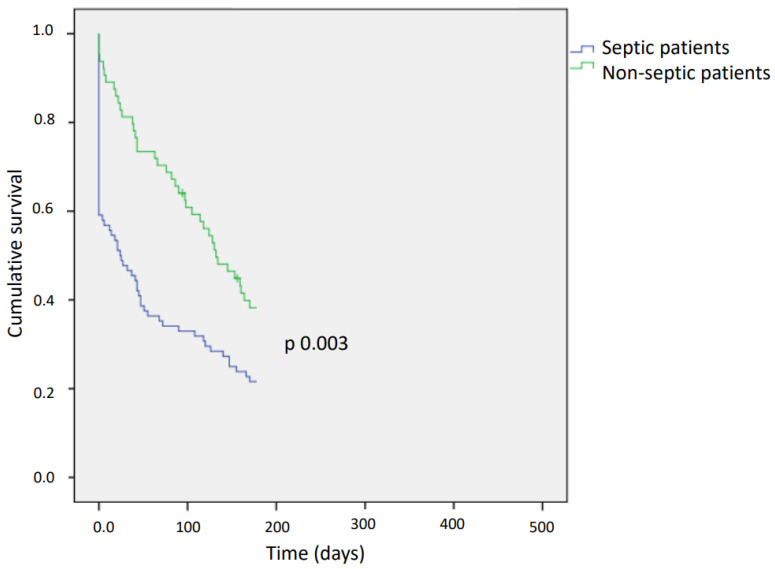
Kaplan–Meier curves of case-fatality rate among patients aged 65 years or older with complicated community-acquired urinary tract infections with and those without sepsis.

**Table 1 microorganisms-11-01959-t001:** Epidemiological and clinical characteristics of complicated community-acquired urinary tract infections in septic and nonseptic patients aged 65 years or older.

	Total N 412	Septic Patients N 197 (47.8%)	Non-Septic Patients N 215 (52.2%)	*p*
Female sex, *n* (%)	209 (50.7)	99 (49.7)	110 (51.2)	0.854
Age (years), median [IQR]	81 [75–88]	83 [77–89]	80 [74–85]	**<0.001**
Charlson ≥ 3, *n* (%)	402 (97.6)	191 (97)	211 (98.1)	0.435
Barthel < 40, *n* (%)	169 (41)	90 (45.7)	79 (36.7)	0.065
**Comorbidities**				
Dementia, *n* (%)	136 (33)	69 (35)	68 (31.4)	0.396
Diabetes mellitus, *n* (%)	161 (39.1)	75 (38.1)	86 (40)	0.689
COPD, *n* (%)	59 (14.3)	25 (12.8)	34 (15.8)	0.377
CKD, *n* (%)	149 (36.2)	71 (36.2)	78 (36.4)	0.962
Cancer, *n* (%)	93 (22.6)	40 (20.3)	53 (24.7)	0.292
Indwelling urinary catheter, *n* (%)	87 (21.1)	39 (19.8)	48 (22.3)	0.530
HCA-UTI, *n* (%)	248 (60.2)	114 (57.9)	134 (62.3)	0.356
Previous hospitalization, *n* (%)	142 (34.5)	69 (35)	73 (34)	0.819
Previous antimicrobial therapy, *n* (%)	212 (51.5)	92 (46.7)	120 (55.8)	0.064
Nursing home residence, *n* (%)	29 (7)	18 (9.1)	11 (5.1)	0.111
**Clinical characteristics**				
APACHE II, median [IQR]	12 [9–16]	16 [12–20]	10 [8–12]	**<0.001**
APN, *n* (%)	262 (63.6)	110 (55.8)	152 (70.7)	**0.002**
Altered mental status, *n* (%)	189 (45.9)	128 (65.3)	61 (28.4)	**<0.001**
RR ≥ 22 bpm, *n* (%)	92 (22.3)	78 (39.8)	14 (6.5)	**<0.001**
SBP < 100 mmHg, *n* (%)	74 (18)	68 (34.7)	6 (2.8)	**<0.001**
Fever, *n* (%)	307 (74.5)	139 (70.6)	168 (78.1)	0.078
qSOFA ≥ 2, *n* (%)	113 (27.4)	109 (55.3)	4 (1.9)	**<0.001**
Septic shock-3, *n* (%)	45 (10.9)	43 (21.8)	2 (0.9)	**<0.001**
Lactate ≥ 2 mg/dL, *n* (%)	171 (41.5)	111 (56.3)	60 (27.9)	**<0.001**
Leukocytosis, median [IQR]	12,900 [9200–17,975]	13,800 [10,100–19,000]	11,800 [8500–16,900]	**0.004**
Blood cultures positive/BC taken, *n* (%)	95/228 (41.7)	60/125 (48)	35/103 (33.9)	**0.001**
MDR-B, *n* (%)	148 (35.9)	70 (35.5)	78 (36.3)	0.875
Polymicrobial UTI, *n* (%)	40 (9.7)	22 (11.2)	18 (8.4)	0.338
IEAT, *n* (%)	107 (26)	48 (24.4)	59 (27.6)	0.460

COPD, chronic obstructive pulmonary disease; CKD, chronic kidney disease; HCA-UTI, healthcare associated-UTI; APN, acute pyelonephritis; RR, respiratory rate; SBP, systolic blood pressure; BC, blood cultures; MDR-B, multidrug-resistant bacteria; IEAT, inadequate empirical antimicrobial therapy. *p* <0.05 is considered statistically significant (in bold).

**Table 2 microorganisms-11-01959-t002:** Outcomes of complicated community-acquired urinary tract infections in septic and nonseptic patients aged 65 years or older. *p* <0.05 is considered statistically significant (in bold).

	Total N 412	Septic Patients N 197 (47.8%)	Non-Septic Patients N 215 (52.2%)	*p*
In-hospital case-fatality rate, *n* (%)	39 (9.5)	36 (18.3)	3 (1.4)	**<0.001**
30-day case-fatality rate, *n* (%)	57 (13.8)	47 (23.9)	10 (4.7)	**<0.001**
6-month case-fatality rate, *n* (%)	107 (26)	67 (34)	40 (18.6)	**0.003**

**Table 3 microorganisms-11-01959-t003:** Univariate and multivariate analysis of risk factors for 6-month case-fatality rate in complicated community-acquired urinary tract infections in patients aged 65 years or older.

	Univariate Analysis		Multivariate Analysis	
	OR (95% CI)	*p*	aOR (95% CI)	*p*
Older than 75 years	3.1 (1.6–5.9)	<0.001	2.5 (1.1–5.6)	**0.026**
Charlson ≥ 3	1.4 (1.1–1.8)	0.048	4.8 (0.5–9.7)	0.999
Barthel ≤ 40	4.1 (2.8–5.9)	<0.001	5.1 (3–8.7)	**<0.001**
Fever	0.6 (0.4–0.8)	0.006	0.5 (0.3–0.9)	**0.018**
HCA-UTI	1.6 (1.1–2.3)	0.008	1.7 (1.1–2.9)	**0.049**
Sepsis (SOFA ≥ 2)	1.6 (1.2–2.4)	<0.001	1.9 (1.1–3.1)	**0.025**
Lactate ≥ 2 mg/dL	1.7 (1.2–2.3)	0.002	1.3 (0.8–2.3)	0.246
IEAT	1.5 (1.1–2.1)	0.019	1.3 (0.7–2.4)	0.500
Bacteremia	1.1 (0.7–1.6)	0.723	1.3 (0.7–2.4)	0.998

HCA-UTI, healthcare-associated urinary tract infection; IEAT, inadequate empirical antimicrobial therapy. *p* <0.05 is considered statistically significant (in bold).

## Data Availability

The data presented in this study are available upon request from the corresponding author.

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
