# Peer review of "Influence of Sepsis on the Middle-Term Outcomes for Urinary Tract Infections in Elderly People"

_microorganisms, 2023, doi:10.3390/microorganisms11081959_

Round 1
Reviewer 1 Report
The following comments are related to the review of the manuscript entitled “INFLUENCE OF SEPSIS ON THE MIDDLE-TERM OUTCOMES FOR URINARY TRACT INFECTIONS IN ELDERLY PEOPLE”. The topic selected has already been discussed in medical literature, but the authors add here the influence of sepsis on the mid-term prognosis. The manuscript is appropriately prepared and the authors used the right methodology to answer their research questions. As there is always room for improvement in a report, following you will find some comments to be considered.
First of all, it has to be said that the outcome studied, mortality (rate), is in fact the case-fatality rate. Mortality rate is defined as the ratio of the number of deaths in the year or selected period of time to the average total population of that year or specified time. However, case-fatality rate is described as the number of deaths in people diagnosed of a certain disease among the number of diagnosed cases, regardless of time or total population; therefore, is a proportion and as such is reported by the authors in the manuscript. Please, contemplate that concept all over the manuscript.
In the Statistical analysis section, use of Cox proportional hazards regression as well as multivariate analysis using logistic regression is stated. As a result, the first technique only produces the statistical significance of the survival curves for septic and non-septic patients. What about median times of survival and risk of death in the 6-month period?
In the Results chapter, please avoid to repeat information: do not duplicate the same results in the text and in the tables. For instance, the paragraph in lines 202-210 includes the same information depicted in Table 2. Once again, review paragraph 218-225 and Table 3 to select the information to be reported in the text and in the Table; moreover, revise data since some figures are different (see OR and 95% CI for Age older than 75 years, among others). By the way, the ORs in multivariate analyses are adjusted OR and must be indicated as aOR. In the table, the right acronym for confidence interval is CI (IC stands for “intervalo de confianza” which is the denomination in Spanish).
Author Response
Reviewer 1
We thank the reviewer for their comments, and below we address each of them, indicating the modifications we have made, which we believe have significantly improved the manuscript.
- First of all, it has to be said that the outcome studied, mortality (rate), is in fact the case-fatality rate. Mortality rate is defined as the ratio of the number of deaths in the year or selected period of time to the average total population of that year or specified time. However, case-fatality rate is described as the number of deaths in people diagnosed of a certain disease among the number of diagnosed cases, regardless of time or total population; therefore, is a proportion and as such is reported by the authors in the manuscript. Please, contemplate that concept all over the manuscript.
Answer: Indeed, the appropriate term in our study is "case-fatality rate" instead of "mortality." Therefore, we appreciate the reviewer's comment, and we have replaced the term "mortality" with "case-fatality rate" throughout the manuscript.
- In the Statistical analysis section, use of Cox proportional hazards regression as well as multivariate analysis using logistic regression is stated. As a result, the first technique only produces the statistical significance of the survival curves for septic and non-septic patients. What about median times of survival and risk of death in the 6-month period?
Answer: Following the reviewer's suggestions, we have calculated the median survival time and the risk of death over a 6-month period. These data have been added to the Results section (lines 241-243).
In the Results chapter, please avoid to repeat information: do not duplicate the same results in the text and in the tables. For instance, the paragraph in lines 202-210 includes the same information depicted in Table 2. Once again, review paragraph 218-225 and Table 3 to select the information to be reported in the text and in the Table; moreover, revise data since some figures are different (see OR and 95% CI for Age older than 75 years, among others). By the way, the ORs in multivariate analyses are adjusted OR and must be indicated as aOR. In the table, the right acronym for confidence interval is CI (IC stands for “intervalo de confianza” which is the denomination in Spanish).
Answer: As suggested by the reviewer, we have amended the text in the Results section to avoid repetition of data in the tables. Specifically, we have changed the text in lines (199-200, 225-226, and 235 in the revised manuscript. These changes have been made taking into account the recommendations of the second reviewer as well). We are grateful to the reviewer for pointing out the error in the text regarding the data cited in Table 3. This has been corrected in the text. The odds ratios in the multivariate analysis have been indicated as adjusted odds ratios (aOR), and the correct acronym for the confidence interval (CI) has been used.
Author Response
Reviewer 2
Thank you very much for your comments, which have contributed to improving the original article. Below, we will respond to each of your comments.
Major concerns:
- The authors address the rationale for the study by affirming that there are no previous studies about the effect of sepsis on medium-term prognosis. However, the Surviving Sepsis Campaign (SSC) recommends that all patients with severe sepsis or septic shock be evaluated as soon as possible for specific sites of infection amenable to source control and undergo source control within 12 hours of diagnosis. This evidence on the impact of source control timing in patients with sepsis is unclear and needs to be further explored considering the existing literature in the introduction.
Following your instructions, a paragraph has been added to the introduction regarding the impact of source control timing in patients with sepsis (lines 46 – 51).
- The structure of the method section has flaws:
- a) Ethics committee approving study and waived the informed consent requirement to the study’s observational should be included in data source and participants (Line 162‑163) with approved protocol number by the Clinical Research Ethics Committee.
Answer: The approved protocol number by the Clinical Research Ethics Committee (code 85/16, September 2016), as well as the following phrase “All patients provided their signed consent to participate in the study” have been added to the manuscript in the Materials and Methods section (Study design and patients), lines 78 - 80.
- b) Does figure 1 belong to the methods or results section? (Line 70 or 166).
Answer: Figure 1 (Flowchart of inclusion of 412 cases of complicated community-acquired urinary tract infection) belongs to the Material and Methods section. Thus, we have removed the reference to figure 1 in line 180 of the Results section.
- c) This prospective observational study needs of additional information about the work that could be in supplementary online content. Examples: Characteristics of all Intensive Care Unit (ICU)-acquired infections after admission for a non-infectious condition, Incidence of ICU-acquired infections over time after admission for sepsis, Admissions at risk at the start per week considering ICU-acquired infections and/or cumulative number of causative pathogens assigned to the ICU-acquired infectious event, since multiple pathogens could be assigned to a single infectious event.
Answer: Due to the study design, only complicated community-acquired UTIs in patients admitted to an Internal Medicine ward were included. Any UTIs that were of nosocomial origin, even if they were community onset, were excluded (n=19, see Figure 1). Therefore, only the episode of UTI that led to hospital admission was analyzed, and in no case were the patients in the ICU.
- d) Why authors did not use intensive care unit term for hospital stay of patients? This term of stay is involved only in the ICU or other unit.
Answer: Patients were in an Internal Medicine ward. They were not in the ICU or other unit.
- e) Values should be given as numbers (%), mean (standard deviation) or median [interquartile range] in statistical analysis section. If there is no data presented in mean (standard deviation), why Student's t-test or analysis of variance (ANOVA) was used? None of the evaluated data had a normal distribution?
Answer: Some of the analysed quantitative variables followed a normal distribution (e.g., RR, respiratory rate; SBP, systolic blood pressure), but when converted to categorical variables, they were expressed as numbers and percentages. Therefore, following the reviewer's instructions, we have removed Student's t-test or analysis of variance from the Materials and Methods section (lines 183-185). None of the evaluated data had a normal distribution.
iii.The results indicate that urinary septic and non-septic elderly patients do not differ for microorganisms isolated in urine culture. This fact suggested a consistent absence of harmful effect of bacteremia on the prognosis of elderly patients evaluated in this study. A possible reason should be presented (line 240), as well as I strongly recommend that the microorganism in acquired infection after admission for non-infectious disease should be included in a table or figure (line 177-191).
Answer: As the reviewer states, there were no differences in microbiological factors, such as microorganisms cultured in urine and blood between septic and non-septic patients. However, it is important to note that in this study, no hospital-acquired infections were included after the initial infection, as all cases corresponded to complicated community-acquired UTIs, and all cases underwent a rigorous selection process (Figure 1).
- The baseline characteristics and outcome of patients admitted with sepsis stratified according to development of UTI-acquired needs be more comprehensible and better separated in the Table 1 (Page 5 and 6) . Examples: Demographics, Chronic comorbidity (none, diabetes, dementia….), Severity of disease (APACHE score, APN…), Treatment interventions (urinary catheter, central venous catheter, mechanical ventilation, corticosteroid use?…)
Answer: This study does not involve patients admitted with sepsis stratified by the development of UTIs. Instead, it focuses on patients admitted with complicated community-acquired UTIs and compares them based on whether they present sepsis or not, in order to understand the medium-term effect of community-acquired urinary sepsis in elderly patients. Therefore, the reviewer's suggestions regarding urinary catheter, central venous catheter, mechanical ventilation, corticosteroid use, etc., do not seem appropriate for this study.
- Data that are not statistically different are repeated numerically in the text. Only difference statistical result could be highlighted. Example: Readers may search on the table the value cited in the lines 168, 169, 193, 203…
Answer: Thank you for the comments. The manuscript has been modified following the reviewer's suggestions (lines 199-200, 225-226, and 235 in the revised manuscript) .
- Antibiotics are prescribed in both primary and secondary care. However, this article did not provide information on this matter ... the manuscript only discussed it briefly about resistance (line 126‑135). Certainly, the antibiotic use changes in the context of antimicrobial resistance, it is very important in the found clinical outcome. This allows a pragmatic approach to assessing the impact of standard care in the community for a large cohort of older patients with confirmed or suspected UTI on adverse events and should be included.
Answer: Thank you for the comments. Following the reviewer’s suggestions, a paragraph has been added to the Results section (lines 252-257).
Minor comments:
- References should be cited in “Consistent with many other previous studies on sepsis from different sources” (line 262, 263).
Answer: Thank you for your comment. Following the reviewer’s suggestion, some references have been added (line 304).
- Figure footers are not complete. The data must be understandable independently of reading the text.
Answer: Thank you for your comment. The legend of Figure 1 has been modified following your recommendation.
Sincerely,
Juan Alberola
Round 2
Reviewer 2 Report
Authors twice cite approval by the clinical research ethics commitee. Please, remove citation from the line 192 until 194 (Statistical analysis section). Just from the line 78, 79, 80 is sufficient.
Also remove percentage of 33 for dementia (line 199).